# Unconventional Rashba Spin-Orbit Coupling on the Charge Conductance and Spin Polarization of a Ferromagnetic/Insulator/Ferromagnetic Rashba Metal Junction

**DOI:** 10.3390/mi13081340

**Published:** 2022-08-18

**Authors:** Aek Jantayod

**Affiliations:** 1Department of Physics, Faculty of Science, Naresuan University, Phitsanulok 65000, Thailand; aekj@nu.ac.th; 2Research Center for Academics in Applied Physics, Faculty of Science, Naresuan University, Phitsanulok 65000, Thailand

**Keywords:** ferromagnetic/insulator/ferromagnetic Rashba metal junction, Rashba spin–orbit coupling, tunneling conductance

## Abstract

A ferromagnetic/insulator/ferromagnetic Rashba metal junction (FM/I/FRM) with both Rashba spin–orbit coupling (RSOC) and exchange energy splitting was studied theoretically. Two kinds of interactions in FRM generate the three metallic states in a FRM; the Rashba ring metal (RRM) state, the anomalous Rashba metal (ARM) state and the normal Rashba metal (NRM) state. The scattering method and the free-electron model are used to describe the transport properties of particles and to calculate the conductance spectrum and the spin polarization of current in the junction. The conductance spectrum in the applied voltage shows the prominent features at the boundaries not only for the three states of the FRM but also in the ARM state. In addition, the conductance in the RRM and ARM states is strongly influenced by both the thickness and barrier height of the insulator layer. We also found that the spin polarization obtains a high value in the ARM state and is not affected by the qualities of the insulator, unlike the RRM and NRM states. Obtaining high-spin polarization from FRM material can be useful to produce spintronic devices in future devices.

## 1. Introduction

Transporting the charge and spin transport in a tunneling junction of a Rashba spin orbit coupling (RSOC) system has promising potential for applications in spin-electronic devices because of the non-symmetrical structure inversion leads to coupling between an electron’s spin and its wave-vector momentum [1,2,3], which can be tuned by the external gate voltage [4,5,6]. Furthermore, spin–orbit coupling with exchange splitting of the magnetization or with an applied external magnetic field are current topics to investigate the spin-electron transport through various junctions [7,8,9,10,11,12], to produce future spintronic devices [13,14,15,16,17,18,19,20,21,22,23].

A material with properties mentioned above is called a ferromagnetic Rashba metal (FRM), which is formed by the coupling between a RSOC and an exchange field, where an inner Fermi surface and outer Fermi surface are separated, generating the three types of metallic states. According to the Fermi level across the band energy spectrum of the FRM, different configurations as well as a different density of states are presented [24,25]. There is the normal Rashba metal (NRM) state, anomalous Rashba metal (ARM) state and Rashba ring metal (RRM) state, where the three metallic states depend on the strength of the RSOC and the exchange splitting of the magnetization [10].

Practically, the difference in the Fermi surface plays an important role to classify the states of the FRM, such as in the NRM state, almost all spin directions are opposite for the two Fermi surfaces. For the ARM state, there is only one outer Fermi surface. In an RRM state, the two Fermi surfaces have almost the same spin directions.

Thin layer heterostructures of Pt/Co/Al oxides [26] were recently reported experimentally in a system with a large RSOC and magnetization, which can be tuned by the gate voltage. Of course, the tunneling conductance of the NM/FRM in a two-dimensional junction has been theoretically studied [24], where the Fermi level of the FRM was turned by the gate voltage to show the three metallic states. The conductance exhibited a unique behavior in the ARM state when the RSOC and the exchange coupling field are taken into account. The states on the strong spin–orbit interaction are usually influenced by both the intrinsic and extrinsic parameters in the heterostructure and show the important electrical features, such as the current, conductance and the spin polarization [27,28,29,30,31,32,33,34,35,36,37].

In this work, we aim to study how the conductance changes with the intrinsic properties of the thin insulator barrier between the FM and the FRM junction. We use the simple technique of the continuous model and the scattering method to study the transport of a particle through the FM/I/FRM junction. The conductance and the spin polarization of the three states of the FRM are investigated by the effect of the square potential, via the insulator between the FM and FRM materials.

The conductance spectrum of the RRM and ARM states are strongly influenced by the thin insulator, as expected and show the characteristic feature change of the conductance spectrum at the boundary of each state. We also calculated the spin polarization of the current dependence with the applied voltage for weak and strong RSOC. The quantum tunneling model and assumptions are introduced in Section 2. The results and discussions are included in Section 3, and the conclusions of this work are in Section 4.

## 2. Model and Assumptions

This model consists of a FM and a FRM with the spin–orbit interaction and the exchange splitting from magnetization by inserting the insulator between the two materials as shown in Figure 1. Our model is an infinite two dimensional system in an xy plane. The scattering method and the free-electron approximation are used to describe the transport properties of a particle through the junction. The free electron Hamiltonian of the system can be express in the three regions as: for x<0 the Hamiltonian of the FM is given by
(1)$${\vec{H}}_{FM}=\frac{{\hat{p}}^{2}}{2m_{FM}}-M_{1}\sigma_{z}-E_{F},$$
where $\hat{p}$ is the momentum operator; $\hat{p}=-i\hbar \left(\hat{x}\frac{\partial }{\partial x}+\hat{y}\frac{\partial }{\partial y}\right)$, $\hat{x}$ and $\hat{y}$ are the unit vector in x- and y- axis, respectively. The Hamiltonian corresponds to the energy spectrum,
(2)$$E_{FM}=\frac{\hbar^{2}{\left(q_{1}^{↑↓}\right)}^{2}}{2m_{FM}}\pm M_{1}-E_{F}.$$
where $m_{FM}$ is the electron mass of FM, $q_{1}^{↑↓}=\sqrt{{\left(q_{1,x}^{↑↓}\right)}^{2}+{\left(q_{1,y}^{↑↓}\right)}^{2}}=\sqrt{2m_{FM}\left(E+E_{F}\mp M_{1}^{2}\right)/\hbar^{2}}$ is the wave vector of the ferromagnetic.

For the middle region; 0<x<L, the Hamiltonian of the insulator is given by
(3)$${\vec{H}}_{I}=\frac{{\hat{p}}^{2}}{2m_{I}}+U_{0},$$
corresponding the energy spectrum—that is,
(4)$$E_{I}=\frac{\hbar^{2}q_{2}^{2}}{2m_{I}}+U_{0}.$$
where $m_{I}$ is the electron effective mass of an insulator and $q_{2}=\sqrt{q_{2,x}^{2}+q_{2,y}^{2}}$ is the wave vector, $q_{2}=\sqrt{2m_{I}\left(E-U_{0}\right)/\hbar^{2}}$ is for $E>U_{0}$, and $q_{2}=Im\left(\sqrt{2m_{I}\left(U_{0}-E\right)/\hbar^{2}}\right)$ is for $E<U_{0}$.

The FRM is in the right side; x>L, it is Hermitian Hamiltonian is [1,10,38,39]
(5)$${\vec{H}}_{FRM}=\frac{{\hat{p}}^{2}}{2m_{R}}+\frac{i}{2\hbar }\left(\hat{z}\cdot \left[\alpha \Theta \left(x-L\right)\left(\hat{p}\times \hat{\sigma }\right)+\left(\hat{p}\times \hat{\sigma }\right)\alpha \Theta \left(x-L\right)\right]\right)-M_{2}\sigma_{z},$$

$m_{R}$ is the electron effective mass of FRM, α is the parameter leading to the strength of the Rashba spin–orbit coupling, $\hat{\sigma }$ is the Pauli spin matrices. The unit vector $\hat{z}$ is direction perpendicular to the xy plane of motion. The Hamiltonian corresponds the energy spectrum—that is,
(6)$$E_{FRM}=\frac{\hbar^{2}{\left(k^{↑↓}\right)}^{2}}{2m_{R}}\pm \sqrt{\alpha^{2}{\left(k^{↑↓}\right)}^{2}+M_{2}^{2}}.$$
where the wave vector is ${\left(k^{↑↓}\right)}^{2}=\sqrt{{\left(k_{x}^{↑↓}\right)}^{2}+k_{y}^{2}}=\frac{2m_{R}}{\hbar^{2}}\left[E+2E_{\alpha }\mp \sqrt{4E_{\alpha }\left(E+E_{\alpha }\right)+M_{2}^{2}}\right]$. The electron effective mass m(x) is position dependent, defined as $m{\left(x\right)}^{-1}=m_{FM}^{-1}\Theta \left(-x\right)+m_{I}^{-1}\left(\Theta \left(x\right)-\Theta \left(x-L\right)\right)+m_{R}^{-1}\Theta \left(x-L\right)$, the Θ(x) is the Heaviside step function, $E_{F}=\left(\hbar^{2}q_{F}^{2}\right)/\left(2m_{FM}\right)$ is the Fermi energy of a ferromagnetic, and the $E_{\alpha }=\alpha^{2}m_{R}/\left(2\hbar^{2}\right)$ is the Rashba energy.

The Rashba spin–orbit interaction divides the energy dispersion relation into two branches, $E_{+}$ and $E_{-}$. When $M_{2}=0$, they come into contact at $k_{x}^{↑↓}=k_{y}=0$, called the crossing point. To include the exchange coupling in the system, the crossing point is split by the magnitude of magnetization $2M_{2}$ as can see in Figure 1. Now, the conduction band becomes the three states, NRM, ARM and RRM states [10,24,25]. Note that the electron with the energy of the plus branch almost presents the up-spin while the minus branch presents the electron with the down-spin—namely, the energy larger than the crossing point (E>0) is the normal Rashba metal state.

The Fermi level is distributed across two bands, $E_{+}$ and $E_{-}$, and there are two Fermi surfaces with opposite spin orientation. Where, the Fermi level passes below the crossing point (E<0), there is only $E_{-}$ band but there is two Fermi surfaces with the same spin direction. An image of the occupied state in the momentum space under this region is the ring shape. Then, the Rashba ring metal (RRM) state is called. When the Rashba spin–orbit coupling and Zeeman splitting are taken into account ($M_{2}\neq 0$), the two bands were split as seen in Equation (Equation 6); note that the $2M_{2}$ is less than $E_{\alpha }$, the only outer Fermi surface is found in $-M_{2}<E<M_{2}$ region.

This is called an anomalous Rashba metal (ARM) state. The width of this state is determined by the $E_{\alpha }$ and $M_{2}$ values mentioned above. In addition, it finds that the interaction between the RSOC and the exchange coupling ( $M_{2}$) causes the minimum energy shift down in the ARM state from E=0—that is, the $E_{c}=\hbar^{2}M_{2}^{2}/2m_{R}\alpha^{2}=M_{2}^{2}/4E_{\alpha }$. In the case of $2M_{2}\geq E_{\alpha }$, the RRM state does not appear because the Zeeman splitting stronger dominates the RSOC, so, the spin orientation on the Fermi surface in the RRM state aligns on the z-direction.

Next, we use the scattering method to write the spin and electron wave functions with the same energy and the same wave vector momentum in y-direction as a linear combination of incident and reflected momentum states. In this work, we injected the electron with up-spin and down-spin states from the FM side through the barrier I and the FRM. The wave functions of an electron with spin in the FM can be written as
(7)$$\begin{array}{c} \psi^{1}{,}_{FM}\left(x,y\right)=\left(\left[\begin{array}{c} 1\\ 0 \end{array}\right]e^{iq_{1,x}^{↑}x}+r_{↑}^{1}\left[\begin{array}{c} 1\\ 0 \end{array}\right]e^{-iq_{1,x}^{↑}x}+r_{↓}^{1}\left[\begin{array}{c} 0\\ 1 \end{array}\right]e^{-iq_{1,x}^{↓}x}\right)e^{iq_{1,y}^{↑}y}, \end{array}$$
(8)$$\begin{array}{c} \psi^{2}{,}_{FM}\left(x,y\right)=\left(\left[\begin{array}{c} 0\\ 1 \end{array}\right]e^{iq_{1,x}^{↓}x}+r_{↑}^{2}\left[\begin{array}{c} 1\\ 0 \end{array}\right]e^{-iq_{1,x}^{↑}x}+r_{↓}^{2}\left[\begin{array}{c} 0\\ 1 \end{array}\right]e^{-iq_{1,x}^{↓}x}\right)e^{iq_{1,y}^{↓}y}, \end{array}$$
where $r_{↑,↓}^{1,2}$ is the reflection coefficient of electron with spin up and spin down.

For the insulator, the wave function is
(9)$$\begin{array}{c} \psi_{I}^{j}\left(x,y\right)=\left(\left[\begin{array}{c} a_{↑}^{j}\\ a_{↓}^{j} \end{array}\right]e^{iq_{2,x}x}+\left[\begin{array}{c} b_{↑}^{j}\\ b_{↓}^{j} \end{array}\right]e^{-iq_{2,x}x}\right)e^{iq_{2,y}y}, \end{array}$$
where j=1,2 is an injection of electron with up-spin and down-spin, respectively, $a_{\sigma }^{j}$ is the transmitted coefficient with spin-σ and $b_{\sigma }^{j}$ is the reflected coefficient with spin-σ in an insulator. Furthermore, the linear combination of the transmission states are written for the wave function of the FRM. The wave function is separated for the two parts; $\psi_{FRM}^{a,j}$ and $\psi_{FRM}^{b,j}$,
(10)$$\psi_{FRM}^{a,j}\left(x,y\right)=\left[\frac{t_{↑}^{j}}{\sqrt{1+\kappa_{↑}^{2}\left(\vec{k}\right)}}\left(\begin{array}{c} g_{↑}\left(\vec{k}\right)\\ 1 \end{array}\right)e^{ik_{x}^{↑}x}+\frac{t_{↓}^{j}}{\sqrt{1+\kappa_{↓}^{2}\left(\vec{k}\right)}}\left(\begin{array}{c} 1\\ g_{↓}\left(\vec{k}\right) \end{array}\right)e^{ik_{x}^{↓}x}\right]e^{ik_{y}y}$$
(11)$$\psi_{FRM}^{b,j}\left(x,y\right)=\left[\frac{t_{↑}^{j}}{\sqrt{1+\kappa_{↑}^{2}\left(\vec{k}\right)}}\left(\begin{array}{c} 1\\ g_{↑}^{*}\left(\vec{k}\right) \end{array}\right)e^{i(-k_{x}^{↑})x}+\frac{t_{↓}^{j}}{\sqrt{1+\kappa_{↓}^{2}\left(\vec{k}\right)}}\left(\begin{array}{c} 1\\ g_{↓}\left(\vec{k}\right) \end{array}\right)e^{ik_{x}^{↓}x}\right]e^{ik_{y}y}$$
where $\psi_{FRM}^{a,j}$ is for $E>-E_{c}$ and $\psi_{FRM}^{b,j}$ is for $E<-E_{c}$; $E_{c}=M_{2}^{2}/4E_{\alpha }$,
(12)$$\kappa_{↑↓}^{2}\left(\vec{k}\right)=\frac{\alpha^{2}{\left(k^{↑↓}\right)}^{2}}{{\left(M_{2}+\sqrt{\alpha^{2}{\left(k^{↑↓}\right)}^{2}+M_{2}^{2}}\right)}^{2}},$$
(13)$$g_{↑↓}\left(\vec{k}\right)=\frac{-\alpha (ik_{x}^{↑↓}\mp k_{y})}{M_{2}+\sqrt{\alpha^{2}{\left(k^{↑↓}\right)}^{2}+M_{2}^{2}}},$$
(14)$$g_{↑}^{*}\left(\vec{k}\right)=\frac{\alpha (ik_{x}^{↑}+k_{y})}{M_{2}+\sqrt{\alpha^{2}{\left(k^{↑}\right)}^{2}+M_{2}^{2}}}.$$

Here, $t_{↑↓}^{j}$, respectively, denotes the transmission coefficients of the plus and minus blanches with its spin, and $q_{1,y}^{↑↓}=q_{2,y}=k_{y}$ (the three states of momentum component along the interface are the same because of the translation symmetry along with one). We can obtain the all of the coefficients by using the boundary conditions at the interfaces,
(15)$$\psi_{FM}^{\left(j\right)}\left(x=0,y\right)=\psi_{I}^{\left(j\right)}\left(x=0,y\right),$$
(16)$$\frac{1}{m_{FM}}\psi_{FM}^{{\left(j\right)}^{\prime }}\left(x=0^{-},y\right)=\frac{1}{m_{I}}\psi_{I}^{{\left(j\right)}^{\prime }}\left(x=0^{+},y\right),$$
(17)$$\psi_{I}^{\left(j\right)}\left(x=L,y\right)=\psi_{FRM}^{j}\left(x=L,y\right),$$
(18)$$\frac{1}{m_{R}}\psi_{FRM}^{{\left(j\right)}^{\prime }}\left(x=L^{+},y\right)-\frac{1}{m_{I}}\psi_{I}^{{\left(j\right)}^{\prime }}\left(x=L^{-},y\right)=-\frac{\alpha }{\hbar }\sigma_{y}\psi_{FRM}^{\left(j\right)}\left(x=L^{+},y\right).$$

Using the conservation of the current density along x-direction, the transmission and reflection probabilities can be calculated. The reflection probability of spin up and spin down is $R_{↑}^{j}={\left|r_{↑}^{j}\right|}^{2}$, $R_{↓}^{j}={\left|r_{↓}^{j}\right|}^{2}$, respectively, and the transmission probability of the plus ($T_{↑}^{j}$) and minus ($T_{↓}^{j}$) branch is $T_{↑↓}^{j}={\left|t_{↑↓}^{j}\right|}^{2}\frac{v_{x,R}^{↑↓}}{v_{x,FM}}$. The group velocity $v_{x}$ can be obtained as $v_{x}\equiv \partial H/\hbar \partial k_{x}$, so, $v_{x,R}^{↑↓}=\frac{\hbar k_{x}^{↑↓}}{m_{R}}\pm \frac{\alpha^{2}k_{x}^{↑↓}}{\sqrt{\alpha^{2}{\left(k^{↑↓}\right)}^{2}+M_{2}^{2}}}$, where the plus and minus sign refer to plus and minus branch of the FRM, respectively, and $v_{x,FM}=\frac{\hbar q_{1,x}^{↑↓}}{m_{FM}}$ [35,39].

Calculating the electric current density *J* across the junction, is given by
(19)$$J\left(V\right)=\frac{eA}{4\pi^{2}}\int_{k_{F,min}}^{k_{F,max}}dk_{y}\int_{0}^{eV}\left(T_{↑}^{j}\left(E,k_{y}\right)+T_{↓}^{j}\left(E,k_{y}\right)\right)\left[f(E-eV)-f(E)\right]dE,$$
where f(E) is a Fermi distribution function, *A* is an area. At zero temperature, the differential conductance defined by G(eV)≡dJ/dV, leads to
(20)$$G\left(eV\right)=\frac{e^{2}A}{4\pi^{2}}\int_{k_{F,min}}^{k_{F,max}}dk_{y}\left(T_{↑}^{j}\left(eV,k_{y}\right)+T_{↓}^{j}\left(eV,k_{y}\right)\right).$$

It should be noted that all conductance in this work are plotted in the units of $\frac{e^{2}}{h}\frac{L^{2}q_{F}}{2\pi }$. Finally, the spin polarization of current *P* in this work is defined by the difference in the current densities with up spin and down spin and can be simply written as
(21)$$P\left(eV\right)=\frac{G_{↑}\left(eV\right)-G_{↓}\left(eV\right)}{G_{↑}\left(eV\right)+G_{↓}\left(eV\right)}.$$

In this section, we explain the methods of calculation to find the equation of conductance spectrum G(eV) as Equation (Equation 18), consequently, to obtain the spin polarization of current P(eV) as Equation (Equation 19). The details for the calculation of the conductance spectra of a junction can be seen in Ref. [37]. An exact solution to them cannot be calculated, and thus the numerical solutions will be presented in the next section.

## 3. Results and Discussion

The main points of discussion in this study are the conductance spectrum (Equation (Equation 18)) and the spin polarization of current (Equation (Equation 19)) of the FM/I/FRM junction, affected by the insulating thickness and the potential barrier height. These two parameters indicate the quality of the junction, which is meant to block the flow of carriers in the off-state condition. Another considered parameter is the electron effective mass, which can be obtained by considering the curvature of the conduction energy dispersion of each material composing the junction. Even though these parameters may not relate to an experiment, we attempt to describe them in terms of their quantitative value.

All the parameters mentioned above can be set up in the real experiment. For instance, the insulator thickness of the magnetic tunnel junction (MTJ) and the potential barrier height of the insulator can be varied, as shown in previous work of MTJ by Saito et al. [40]. The strength of Rashba spin–orbit coupling and the exchange splitting energy used in this work may not entirely match the real value of the experiment. However, Rashba spin–orbit coupling parameter presented either interface or bulk inversion symmetry breaking [41,42] can be tuned by the gate voltage, and its strength is not limited [4,43,44].

Similarly, the exchange coupling is proportional to the magnetization, which can be manipulated by using an applied external magnetic field. In this paper, the heterostructure of FM/I/FRM comprising two types of spin coupling states represent the characteristic feature of the conductance and the spin polarization, which is an interesting alternative for the experiment of the novel spintronic devices.

First, we discuss the conductance spectrum (*G*) of the FM/FRM junction as a function of the applied bias voltage (eV) for various ratios of the effective mass ($m_{R}/m_{FM}$), we let $m_{I}=m_{FM}$. Figure 2a displays the normal RSOC (without exchange coupling energy; M=0). The *G* depends on eV as it rapidly increases when the bias voltage reaches the band bottom of the FRM and it monotonically enhances with increasing voltage. When the $m_{R}/m_{FM}$ decreases, the *G* is suppressed, which can be attributed to the band mismatch of the conductivity between the FM and FRM materials.

Figure 2b,c shows the *G* that includes both RSOC and magnetization as assigned by $0<M_{2}<E_{\alpha };M_{2}=0.2E_{F},E_{\alpha }=0.4E_{F}$ and $M_{2}>E_{\alpha };M_{2}=0.4E_{F},E_{\alpha }=0.2E_{F}$, respectively. These coexisting of RSOC and magnetization lead to unconventional Rashba effect. For the exchange coupling energy less than the Rashba energy ($0<M_{2}<E_{\alpha }$), Figure 2b, the energy dispersion relation of FRM is split for $2M_{2}$. As a result, the Fermi surfaces of FRM are separated into three states: RRM, ARM and NRM states, as can be seen in FRM region (in Figure 1).

This situation causes the distinguishing characteristic features of the *G* for the three states as shown in Figure 2b. For $M_{2}$ is larger than $E_{\alpha }$, a strong coupling exchange of the magnetic properties leads to the two states of ARM and NRM as presented in Figure 2c. In the mean time, the *G* is also suppressed by decreasing the ratio of $m_{R}/m_{FM}$. Similar behavior of *G* curve has also been found in previous works of M/FM junction [45,46]. According to the conductance spectrum of the metal/2DEG with a RSOC junction, the Rashba energy can be predicted directly from the characteristic features of the *G* vs. eV curves [35,36,37]. In this paper, the *G* dependence on the applied voltage in a FM/FRM junction shows interesting features needed to determine the type of Rashba effects in the FRM, such as RRM, ARM and NRM states.

Second, Figure 3 shows the *G* vs. eV of FM/I/FRM junction in case of $0<M_{2}<E_{\alpha }$ for different values of the potential barrier height; $U_{0}$. With the Rashba energy and exchange coupling energy set as $E_{\alpha }=0.4/E_{F}$, and $M_{2}=0.2E_{F}$, respectively, the thickness of insulating layer ($Lk_{F}$) is set as 1.0 (Figure 3a) and 10.0 (Figure 3b). It was found that when the thickness of insulator has a small value (Figure 3a), the *G* is suppressed with increasing $U_{0}$.

Normally, a thin thickness of insulator acts as a Dirac delta potential between two materials, and a high barrier can block the particle crossing the junction as well [47]. Interestingly, the kink of *G* is displayed at the $eV=E_{C}$ in the ARM state, unlike in charge conductance of conventional Rashba system junction [35,45]. The kink shows in the ARM state because an evanescent wave occurs; ${\left(k_{x}^{↑}\right)}^{2}<0$. The energy band $E_{+}$ and $E_{-}$ meet at $eV=-E_{c}$, and ${\left(k^{↑}\right)}^{2}$ is given by $E_{+}$ for $eV>-E_{c}$ and is given by $E_{-}$ for $eV<-E_{c}$. In other words, an evanescent wave corresponding to $k^{↑}$ is used by the two branches under conditions; ${\left(k_{x}^{↑}\right)}^{2}<0$.

A discontinuity in the conductance spectrum is presented. However, the prominent kinks of *G* are reduced when the barrier height and the thickness of the insulator have a large value (see Figure 3b). Furthermore, at an applied voltage below $E_{c}$ ($eV<-E_{c}$), it can be clearly seen that the *G* depends strongly on $U_{0}$ more than any other region ($eV>-E_{c}$). On the other hand, for $Lk_{F}=10.0$ (Figure 3b), for $eV<-E_{c}$, the *G* does not monotonically decrease with increasing $U_{0}$.

Instead, it exhibits a maximum value at the set up value of $U_{0}$. Basically, the thickness of barrier between two regions causes the oscillation of particle across such junction. The periods of the oscillation increase with increasing the thickness ($Lk_{F}$). The peaks of the oscillation directly refer to the energy level of the particle through the junction. Thus, the *G* is sensitive to the interfacial barrier and thickness of insulator of such junction.

In addition, the *G* in case of $M_{2}>E_{\alpha }$ was also investigated. When the thickness was small ($Lk_{F}=1.0$, Figure 4a), the *G* was normally suppressed by increasing the $U_{0}$ in both the ARM and NRM states. In contrast, as thickness increases ($Lk_{F}=10.0$, Figure 4b), the *G* increases when the $U_{0}$ is raised to $U_{0}$ = 2.0$E_{F}$ and then gradually decreases by further raising of $U_{0}$ to 3.0 and 5.0$E_{F}$. Furthermore, the anomaly change in *G* occurs at $eV=E_{c}$ in the ARM state. Interestingly, the strength of Rashba spin–orbit coupling and the exchange energy splitting in FRM can be directly determined from the spacing between the onset and the changing of the slope of *G* (i.e., slope anomaly) as well as the states. Again, the thickness and the barrier of the insulator need to be carefully controlled to achieve a maximum value of *G*.

Generally, two materials are mismatch conductivity at the interface. It means that there is a potential between them. Simplicity, it can describe this potential by using Dirac delta function [48] at the interface in such junction; for example, Zδ(x), *Z* is the strength of the barrier height. It can be varied by parameter Z. Normally, the electrical current (*I*) vs. applied voltage (eV) suppresses when the parameter *Z* increases. Since the high potential strength can block the particle tunneling the junction. In the experiment, a thin insulator is a famous to insert between two materials. It finds that the a insulator can reduce the conductivity mismatch between a junction.

However, the thickness of insulator is not much. If the thickness of insulator is too thin, it acts like a considering interfacial barrier with the Dirac delta potential [47]. It seen that, the heterostructure is significantly on the quantitative observation. The thickness of insulator is one important material to affect the transport properties of the heterostructure of FM/I/FRM, the particle with its spin travels a long distance in an insulator between FM and FRM. The spin relaxation effect can occur depending on their structure, and this effect may not be seen in the total conductance. For a thin insulator case, it looks like a delta potential between the FM and FRM. The conductance spectrum is also look similar with the metal/FRM junction in Ref. [25].

On the other hand, when a thicker insulator and increasing the barrier potential strength are taken into consideration (Figure 3b and Figure 4b), the *G* in ARM region obviously change non-monotonic behavior. The *G* obtains a high value under conditions. In the experiment, the optimal layer thickness depends significantly on the electrodes in addition to other parameters, such as the deposition method, rate and the nature of the material [49,50]. Additionally, the spin relaxation of electrons traveling long distances between electrodes (thick insulator) results in a reduction of the kink at $eV=E_{c}$. Inversely, the configuration of this kink can roughly estimate the thickness of the insulating layer in such a junction or magnetic tunnel junction in terms of the thickness or thinness.

Next, the spin polarization of current as a function of the applied voltage defined in Equation (Equation 19) is considered. Various potential barrier heights and thicknesses of insulator were also studied to investigate the effect on the spin polarization. Figure 5a is for $Lk_{F}=1.0$, and Figure 5b is for $Lk_{F}=10.0$. The parameters $E_{\alpha }$ and $M_{2}$ are set to the same value as in Figure 3. In the case of $0<M_{2}<E_{\alpha }$, the three states of FRM are presented. With increasing applied voltage, the spin polarization first increases in the RRM state and reaches 100% in the ARM state before continuously decreasing in the NRM state.

When the insulator thickness is either set as $Lk_{F}=1.0$ or $Lk_{F}=10.0$, it was found that increasing the barrier height has a small effect on spin polarization. Moreover, the *P* in the ARM state is not affected when either the thickness or the barrier height of the insulator are changed. It is because the ARM state is found only at one outer Fermi surface, which contains one spin direction, leading to high-spin polarization across the junction.

Figure 6 displays the *P* vs. eV relationship in case of $M_{2}>E_{\alpha }$. There are two states represented, ARM and NRM states. The parameters are set to be the same as those of Figure 4. The *P* in ARM state for either $Lk_{F}=1.0$ or $Lk_{F}=10.0$ is independent of the $U_{0}$ and the thickness of the insulator, whereas the *P* in the NRM state slightly increases with increasing $U_{0}$. Since the energy dispersion as well as the Fermi surface of FRM of the case $M_{2}>E_{\alpha }$ are similar to the FM conduction band for low energy, the exchange splitting dominates the RSOC; therefore, the outer Fermi surface carries a majority spin, resulting in 100% spin polarization.

It is known that the energy dispersion relation of the Rashba system is split into two-subbands. Each band does not have a specific direction of electron with spin-up or spin-down. Despite the fact that a Fermi surface below the crossing point (E=0) has only one outer surface, spin polarization cannot be measured perfectly in theory [35,45]. In novel spintronic devices, such as spin field effect transistors, the control of spin-polarized injection from a ferromagnetic metal into semiconductors with a Rashba spin orbit coupling system is partially attractive for the realization of the device concept. The non-zero incident angle injection of electrons in the left non-magnetic material into the Rashba system plays a crucial role in obtaining a significant spin polarization of current, which is an alternative way as proposed by ref. [51].

Furthermore, manipulating the incident electron energy of a two-dimension with both Rashba and Dresselhaus spin–orbit interaction was accomplished in previous work by ref. [52]. In their study, the very high degree of spin polarized injection into refraction medium is obtained and it may be easier to experimentally achieved than the adjusting angle of electron incidence. In this paper, the 100% spin polarization can be obtained in the ARM state for both strong Rashba spin orbit coupling and strong exchange energy splitting by injecting the electron incident form the FM/I/FRM heterostructure.

## 4. Conclusions

We studied the heterostructure of a FM/I/FRM interface with both the RSOC and the exchange splitting in an electric field and calculated the conductance spectrum of an electron for applied voltages via the free particle approximation and the scattering method at absolute zero temperature. We found that the conductance spectrum for the applied voltage displayed significant characteristic features at the boundaries of the RRM, ARM and NRM states and also in the ARM state. The barrier height and thickness of the insulator need to be carefully adjusted in the junction to find a suitable value for the conductance spectrum in each state of the FRM. Particularly in the RRM state, the conductance spectrum depicts a non-monotonic curve by changing either the barrier height or the thickness of the insulator. Furthermore, the ARM state has only one outer Fermi surface, resulting in high-spin polarization. This is a major key to producing spintronic devices in the future.

## Figures and Tables

**Figure 1 micromachines-13-01340-f001:**
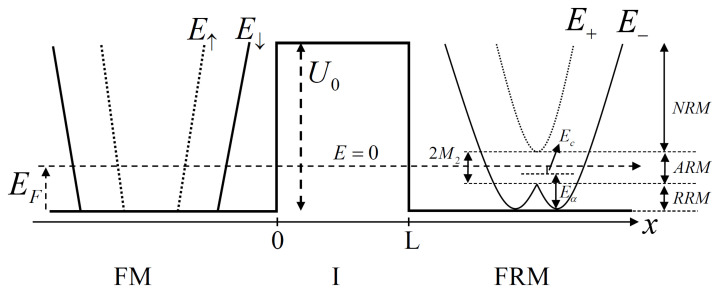
A schematic diagram of the energy spectra of electron in FM/I/FRM junction, the parameters were displayed in the text.

**Figure 2 micromachines-13-01340-f002:**
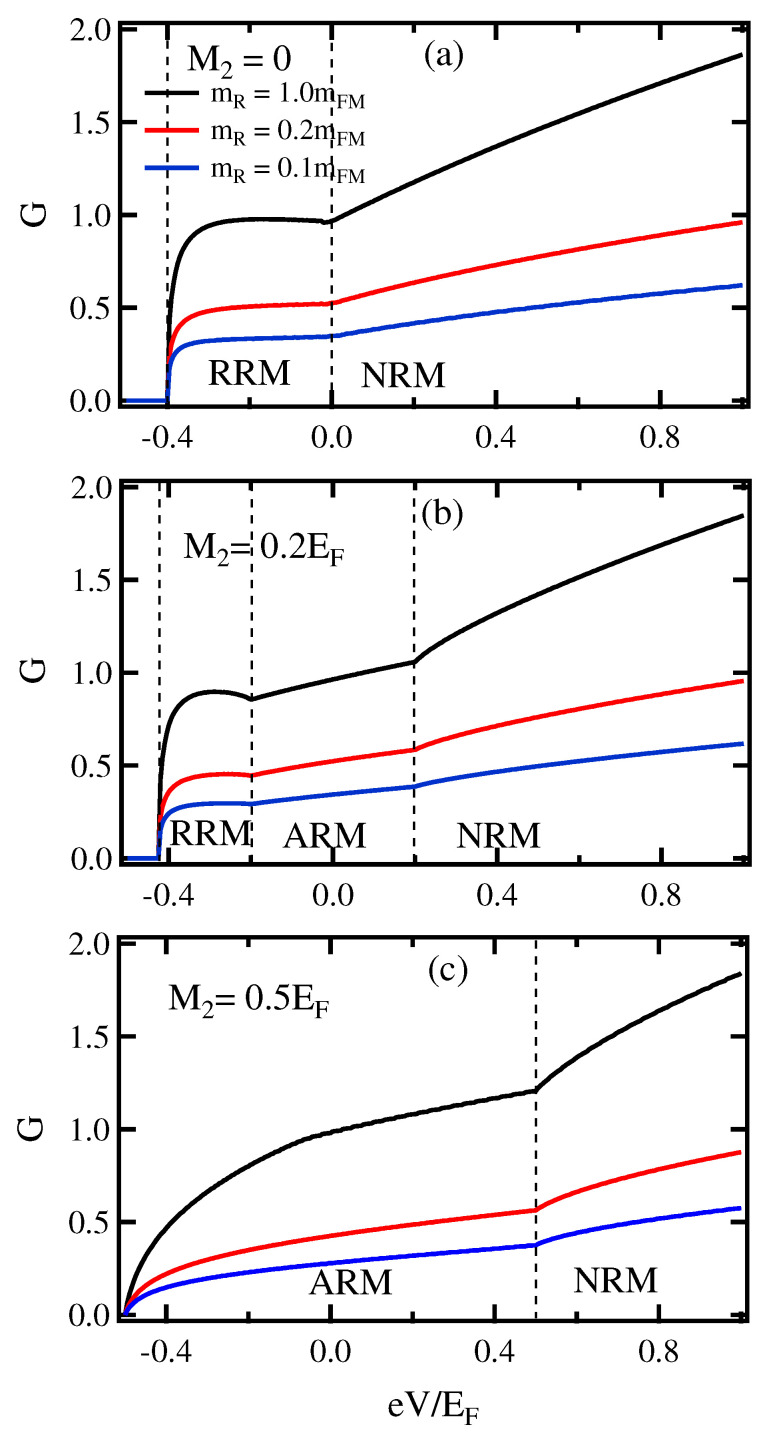
(Color online) Conductance spectra in the applied voltage for different values of the electron effective mass, the barrier potential $U_{0}=0$. (**a**) the normal RSOC; $M_{2}=0$, (**b**) $M_{2}<E_{\alpha }$, (**c**) $M_{2}>E_{\alpha }$.

**Figure 3 micromachines-13-01340-f003:**
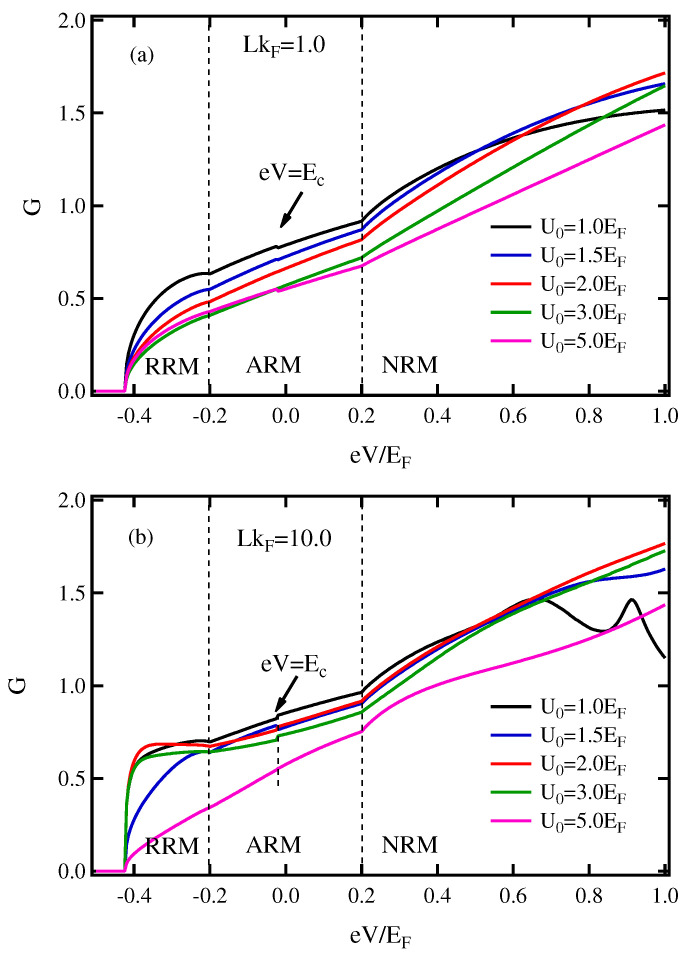
(Color online) Conductance spectra in the applied voltage for different values of the barrier potential in case of $0<M_{2}<E_{\alpha }$. (**a**) $Lk_{F}=1.0$, (**b**) $Lk_{F}=10.0$. Here, the parameters were set as $m_{R}=m_{FM}$, $M_{2}=0.2E_{F}$ and $E_{\alpha }=0.4E_{F}$.

**Figure 4 micromachines-13-01340-f004:**
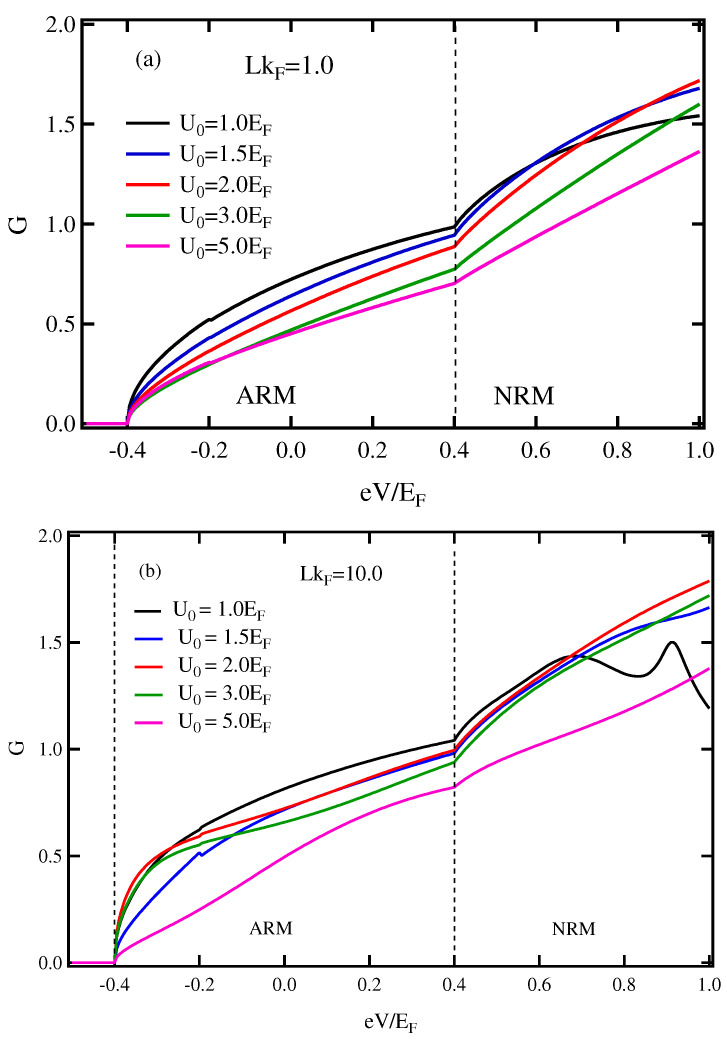
(Color online) Conductance spectra in the applied voltage for different values of the barrier potential in case of $E_{\alpha }<M_{2}$. (**a**) $Lk_{F}=1.0$, (**b**) $Lk_{F}=10.0$. Here, the parameters were set as $m_{R}=m_{FM}$, $M_{2}=0.4E_{F}$ and $E_{\alpha }=0.2E_{F}$.

**Figure 5 micromachines-13-01340-f005:**
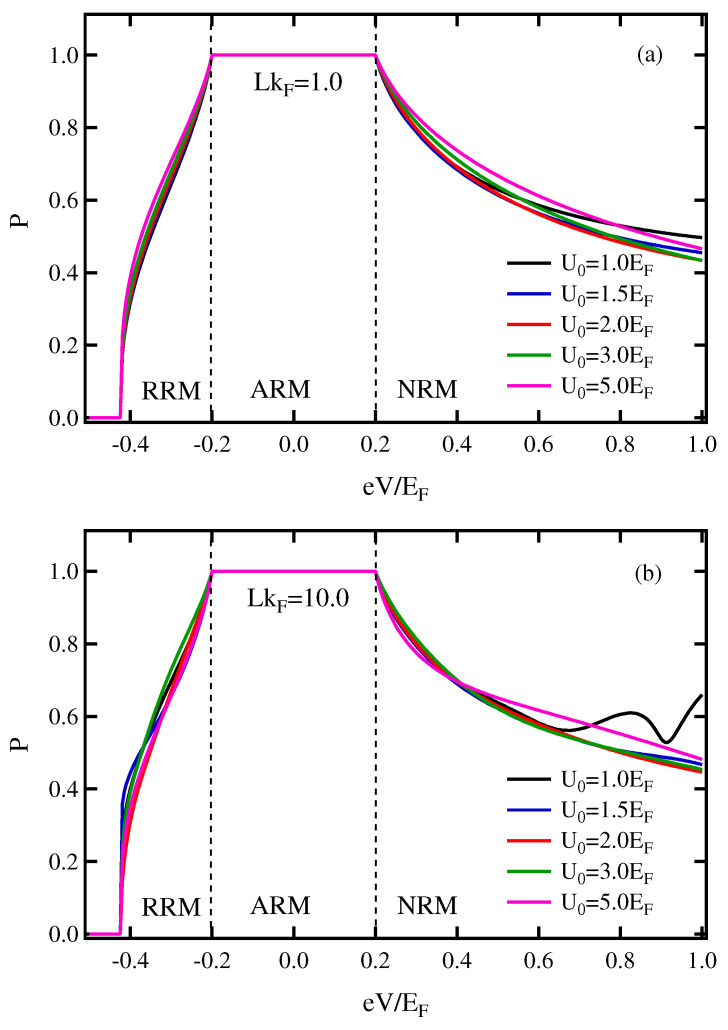
(Color online) Conductance spectra in the applied voltage for different values of the barrier potential in case of $0<M_{2}<E_{\alpha }$. (**a**) $Lk_{F}=1.0$, (**b**) $Lk_{F}=10.0$. The parameters were set as the same in Figure 3.

**Figure 6 micromachines-13-01340-f006:**
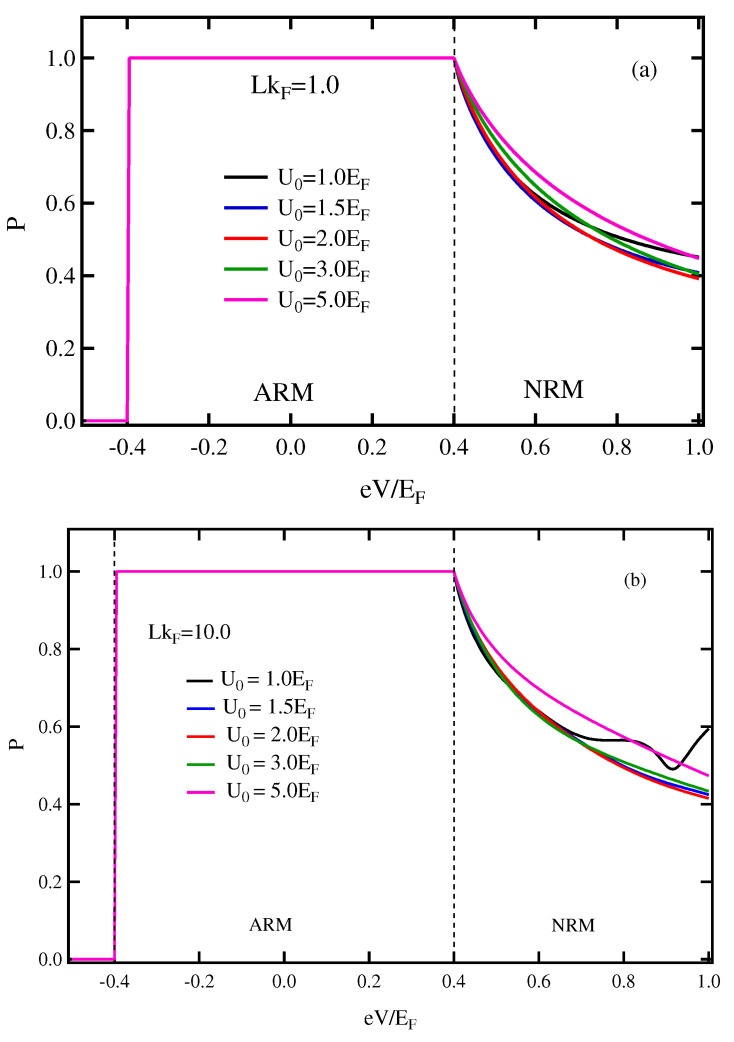
(Color online) Spin polarization of current in the applied voltage for different values of the barrier potential in case of $M_{2}>E_{\alpha }$. (**a**) $Lk_{F}=1.0$, (**b**) $Lk_{F}=10.0$. The parameters were set as the same in Figure 4.

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
