# Peer review of "Unconventional Rashba Spin-Orbit Coupling on the Charge Conductance and Spin Polarization of a Ferromagnetic/Insulator/Ferromagnetic Rashba Metal Junction"

_micromachines, 2022, doi:10.3390/mi13081340_

Round 1
Reviewer 1 Report
The author has theoretically studied the conductance spectrum and the spin polarization of current in the ferromagnetic/insulator/ferromagnetic Rashba metal junction and obtained some interesting results. I suggest this work to be published in Micromachines after some improvement.
1. It seems junction is metallic, the author only calculated the conductance properties contributed by the three special metallic states. Is there any other states near Fermi level to affect the conductance properties? Is any realistic materials to support your model?
2. In the conductance spectra, there are many unusual changes, such as the G at about 0.5 in Fig2(c). Can the author give more explanation about such type of changes?
3. Are these results dependent on the materials or interface effects in junction? For example, if we use different materials to construct the junction, these results are still valid?
4. Also the language should be improved.
Author Response
Dear Reviewer
The responses to your comments can see in the attached file.
Thank you for your reminders that helped us improve this manuscript and new work in the future.
Aek Jantayod

Reviewer 2 Report
The author studies the two-dimensional ferromagnetic metal/insulator/ferromagnetic metal with Rashba spin-orbit coupling junction. It seems that this paper lacks the definition of the wave vector and contains many mistakes. Therefore it is hard for me to judge whether the results are correct or not. Also, the author points out there is a kink in the conductance at eV=E_c but the author does not discuss the origin of it. I recommend the author to check that the obtained results are reproduced from other methods. For example, lattice model ore something. Without such kind of confirmation, I cannot recommend the publication.
1. In Eqs. (1), (3), and (5), m_{FM}(x), m_{I}(x), and m_{R}(x), respectively, should be replaced by m(x) or remove "(x)".
2. In the definition of momentum operator p, there are \hat{x} and \hat{y}. It is better to give the definition of them.
3. In Eq. (5), there are k_x and k_y. They should be derivative operators.
4. In Eq. (6), there is \pm in the square root. It must be +.
5. In the definition of m(x), corresponding to m_{I}, \Theta(x)+\Theta(L-x) --> \Theta(x)\Theta(L-x): "+" should be removed or does the author use this definition?
6. Below Eq. (6), there is a sentence "When M_2=0, they contact at k = 0". "k" should be replaced with "k_{\uparrow\downarrow}". Do I understand correctly? Please define "k".
7. Below Eq. (6), there is a sentence "Namely, the two splitting bands of a normal Rashba system meet at zero wave vector momentum (k = 0) as well as the energy level is zero. It is called the crossing point.". But it is already explained in the same paragraph. What does the author want to explain here?
8. In, Eq. (7), the wave vector of the third term should be q^{\downarrow}_{1,x}. Eq. (8) is the same that the second and the third term contain the same wave vector but they should be different.
9. In Eqs. (15) and (16), I could not understand the boundary condition of the derivative of the wave functions. For example, FM is x<0 but there is \psi'_{FM}(0^+). What does this mean? \psi_{FM} is defined in the entire region?
10. In Eqs. (10) and (11), k_{x}^{\uparrow} and k_{x}^{\downarrow} are not defined. Also q_{1,\uparrow}, q_{1,\downarrow}, and q_{2,x} are not defined.
11. In Eqs. (13) and (14), there is an additional bracket in the denominator.
12. In Eq. (9), the y dependence is given by e^{q_{2,y}y}. If q_{2,y} is a real positive, the wave function diverges for y-->+\infty. This wave function seems to be unphysical.
13. Below Eq. (11), there is a sentence that Eq. (10) should be chosen for E>-E_c and Eq. (11) should be chosen for E<-E_c. On the other hand, for -M_2<E<M_2, the k_x for the \up branch is imaginary and the wave function must be an Evanescent wave. Without giving the definition of k_{x}^{\uparrow}, I cannot judge whether the wave functions are correct or not and I am not sure that the results for the ARM region is correct or not.
14. On page five, last paragraph, there is an open bracket that does not close: (0<M_2<E_\alpha, ...
15. Below Eq. (6), there is a definition of the Rashba energy. In the definition, m_{FR} --> m_{R}.
16. In Fig. 2(c) "=0.5E_{F}" should not be an index.
17. In Fig. 6, labels "(a)" and "(b)" are missing. Also, in the second graph, "U" should be "U_0".
Author Response

(The authors gave the same response as above.)

Reviewer 3 Report
In this manuscript the author addresses the transport properties of the junction between the ferromagnetic electrodes adding the Rashba spin-orbit coupling in one of the electrodes. They calculate the conductance and spin polarization of the current using the scattering approach. The resulting curves reveal the peculiarities caused by the spin splitting of the electronic spectrum. These peculiarities are very much similar to the ones found in Refs.24-25. Also considering the physics of the effects under study one can see that the key effects and mechanisms of new transport features in this manuscript and Refs.24-25 are similar. Though, of course, there is some difference in particular systems studied in these papers (in this paper the author considers two ferromagnetic electrodes while the authors of [24-25] take the junction between the ferromagnetic electrode and normal metal). Thus, I can not recommend the paper for publication because of the lack of novelty. Probably the paper could be improved after major revision and including some additional results and new physics compared to previous publications.
Author Response

(The authors gave the same response as above.)

Round 2
Reviewer 1 Report
The authors have addressed all comments, so I recommend accepting the manuscript.
Author Response
Thank you very much for all comments that helped us improve this manuscript and a new work in the future.
Reviewer 2 Report
The paper improved a lot, but there are still several comments and questions. Therefore, I am still unsure whether the calculation is correct.
1. Below Eq. (1), there are \hat{x} and \hat{y}. It is better to give the definitions.
2. In Eq. (2), the index for q is missing: q^{\uparrow\downarrow} --> q_{1}^{\uparrow\downarrow}.
3. Below Eq. (5), there is "k_x". The definition of "k_x" is not given. As I commented previously, in Eq. (5), the wave numbers should be replaced with the derivative operators, and the corresponding boundary condition (The second equation in Eq. (16)) should be changed. If I understand correctly, the corresponding term in the Hamiltonian is given by (\sigma_x k_y-\sigma_y k_x) --> -0.5i*[\Theta(x-L)(\sigma_x \partial_y - \sigma_y \partial_x)+(\sigma_x \partial_y - \sigma_y \partial_x)\Theta(x-L)]. Here, there are two terms since the Hamiltonian is hermitian.
4. Below Eq. (16), the definition of T_{\uparrow\downarrow} is given. On the right-hand side of this definition, the superscript "j" remains. Does the author fix j to calculate conductance?
Author Response
Dear reviewer
Thank you for your reminders. We revised the manuscript as your comments in the attached file.
Sincerely yours
Jantayod Aek

Reviewer 3 Report
I agree with the remark of the author that the result concerning the spin polarization of the current looks new. At the same time I completely do not understand the note regarding the critisism of the model with the delta-barrier: "However, the potential barrier in Refs. 24-25 used the Dirac delta function potential which only limits the barrier's strength to varying degrees, tends to decrease with increasing the potential strength. In realistic model or in the experiment set up generally insert by a thin insulator between the materials. The thickness also strongly depends on the quality of the junction. As a result, in our junction, when the thickness is increased, the conductance changes with non-monotonic behavior. So, both the barrier height and the thickness of a thin insulator play a role in considering the quality of the heterostructure."
The strength of the delta function potential depends both on the height and thickness of the barrier. If the authors in their calculations observe some qualitative deviations from the results of the delta barrier model I would ask them to describe and explain this descrepancy in detail. If the deviations are only quantitative then the results still look not new enough.
Author Response

(The authors gave the same response as above.)

Round 3
Reviewer 2 Report
I cannot entirely agree with Eq. (5). The left-hand side is an operator. The first term of the right-hand side is an operator, but the second term is not. I would like to hear the author's opinions against following six comments and questions.
1. If "p" in the second term is a c-number, the second term in the right-hand side in Eq. (5) equals to zero. Then, the Hamiltonian does not include spin-orbit interaction. If so, for example, Eq. (6) is incorrect that it must not include \alpha.
2. I think, "p" in the second term must be the operator. After acting on a wave function, it can be replaced with a c-number.
3. Then, because of the presence of the second term which is the operator, the second boundary condition in Eq. (16) must include the first and zeroth derivative of \Phi_{FRI}.
4. Also, I think "-" in the second term in Eq. (5) should be replaced with "+".
5. Below Eq. (5), the author give the definition of "p" that it has two components. If it has only two components, how the author define outer product between \sigma and p?
6. In the second term in Eq. (5), there are outer product of \sigma and p. It seems that the author picks up z-component of the outer product, or "\times" in the second term is not the outer product? Please give the definition of "\times" in the second term.
Minor comment:
The left-hand side of Eq. (17) should depend on "j" because the right-hand side depends on "j". The same thing occurs in Eq. (18).
Author Response
Dear reviewer
Thank you for your reminders. We revised the manuscript as your comments in the attached file.
Sincerely yours

Reviewer 3 Report
I see that the author is trying to explain the difference between the results of his work and the ones published previously. I am almost ready to agree with his line of reasoning. Still I strongly recommend to include all the details of this discussion not only in his reply to my comments but also in the text of the manuscript.
Author Response
Dear reviewer
Thank you for your reminders. We add the details in the lines 187-211 and references 43-44 in the update manuscript.
Sincerely yours